# Leveraging faith-leaders to prevent violence against women and girls: A qualitative study of evangelical faith-leaders' perceptions in Woliso, Ethiopia

**Wosene Berhanu**[1☯‡], **Elisa Gobbo**[2*☯‡], **Addise Amado Dube**[3], **Nesanet Megersa**[1], **Yoska Amenu**[4], **Mengistu Demeke**[3], **Adamu Addisse**[5], **Sibylle Herzig Van Wees**[2]

**1** Department of Development Studies, Ethiopian Graduate School of Theology, Addis Ababa, Ethiopia, **2** Department of Global Public Health, Karolinska Institute, Solna, Sweden, **3** Department of Theology and Biblical Studies, Ethiopian Graduate School of Theology, Addis Ababa, Ethiopia, **4** Department of Development Studies, Evangelical Theological College, Addis Ababa, Ethiopia, **5** School of Public Health, Addis Ababa University, Addis Ababa, Ethiopia

☯ These authors contributed equally to this work.
‡ These authors share first authorship on this work.
* elisa.gobbo@ki.se

## Abstract

### Background

Violence against women and girls (VAWG) affects 1 in 3 women globally. Religion can be both a driver and a potential solution for VAWG. Evidence suggests that faith-leaders can create positive change around VAWG and improve ideas of gender equality, but it is critical to have a context-specific understanding of existing awareness. In Ethiopia, Evangelical faith-leaders' perceptions of VAWG have not been studied.

### Aim

To increase understanding of Evangelical faith-leaders' perceptions of gender equality and VAWG and their views on how they can address VAWG, in Woliso, Ethiopia.

### Methods

A qualitative study design drawing on 14 semi-structured interviews and observational data from workshops with Evangelical faith-leaders that explored their views on gender equality and VAWG. An inductive reflexive thematic analysis approach was used.

### Results

The data analysis generated 4 themes, 8 sub-themes, and 22 categories. The themes include: (1) differing perceptions of VAWG as a problem in the community; (2) navigating between social and religious norms as explanations of VAWG; (3) differing interpretations of biblical scripture; and (4) navigating who has a role in countering gender inequality and VAWG. Many faith-leaders condemned forms of VAWG and expressed hopes for equality,

**Data availability statement:** The datasets generated and/or analysed during the current study are available on reasonable request. Please contact Ruth Mekuria at the Ethiopian Graduate school of Theology for access or information regarding the data and materials ruthmekuria@egst.edu.et.

**Funding:** This work was supported by the New Investigator Award from Karolinska Institutet (2022-0016 to SHvW). The funders had no role in study design, data collection and analysis, decision to publish, or preparation of the manuscript.

**Competing interests:** The authors declare that they have no competing interests.

but some failed to grasp all elements of VAWG and utilized religious reasoning to justify VAWG. The majority of faith-leaders expressed an interest in increasing awareness and held hopes for improved gender equality.

## Conclusion

Faith-leaders hold different understandings of what constitutes VAWG. Faith-leaders' hopes for equality and their willingness to address VAWG indicates the potential of faith-leaders as important actors in VAWG prevention efforts. Yet, trainings and support are critical.

## 1 Introduction

### 1.1 Violence against women and girls

Violence against women and girls (VAWG) is a global problem affecting approximately 1 in 3 women [1]. The World Health Organization (WHO) defines VAWG as "any act of gender-based violence that results in, or is likely to result in, physical, sexual, or mental harm or suffering to women, including threats of such acts, coercion or arbitrary deprivation of liberty, whether occurring in public or in private life" [1]. The most common form of VAWG is intimate partner violence (IPV), which is violence perpetrated by a current or previous boyfriend, husband, or partner [2].

Despite efforts made by numerous groups and institutions, VAWG remains widespread [3]. The implications of VAWG are multifaceted ranging from negative effects on mental health and wellbeing to risk of death [1,4]. The causes of VAWG are complex and vary in different contexts. Yet, patriarchal norms, described as a system of society where men hold power, and women are largely excluded [5], and gender inequalities are influential societal-level factors associated with VAWG [6]. For example, a qualitative study among perpetrators of VAWG in Ghana concluded that social and cultural factors rooted in patriarchy generated ideas of traditional masculinity (defined as men as decision makers, clearly defined gender roles, and men's power over women) that lead to violence [7]. The concept of traditional masculinity is complex and varies across time and culture but has been defined and measured as a prescriptive and gender-specific idea that includes conceptions of self-reliance, toughness, restrictive emotionality, avoidance of participating in feminine activities, importance of having sex, dominance, pursuit of status, and men being the main decision-makers [8–10].

For this research, the term gender refers to economic, social, and cultural factors associated with being a man or woman. Additionally, in this study, gender equality refers to the notion that men and women have the same access to opportunities and life changes and that they are not dependent on or constrained by their gender. Thus, to achieve gender equality, it is necessary to identify and address imbalances based on social and historical injustices through gender equity. Gender equity is the process of being fair to both genders based on their needs, and often entails adjusting for disadvantages faced by women [11].

### 1.2 Religion and VAWG

In addition to patriarchy, religion influences perspectives on gender norms and plays a role in both the perpetuation and prevention of VAWG. Religion is defined as both a normative order linked with a set of practices and the belief in other realities, which combines the substantive and functional definitions [6,12]. Faith and religion are colloquially used interchangeably, but for the purpose of this paper "faith" is defined as the human belief in a transcendent reality,

and "religion" as the institutionalized system of beliefs [13]. Here, the term religion will be utilized to explain norms and shared beliefs. The term faith-leaders will be used to describe the leaders from the Evangelical religious institutions in Ethiopia. Using this term allows for continuity within the field [14].

The study presented here focuses on religion as a social and cultural phenomenon rather than from a theological perspective [6,15]. Examining it from the social science perspective concentrates on the implications of religious beliefs and organizations on society, rather than attempting to provide a rationale for religious ideas [15]. Through this social and cultural scope, religion can be examined as both a driver and a solution for VAWG [6].

Religious ideas, practices, and organizations can be used to support patriarchal ideas, legitimize abusive behaviors, and promote silence [6]. However, religion can also promote ideas of gender equality, empathy, and kindness, which can aid in VAWG prevention efforts and contribute to supportive environments for VAWG victims [6]. Thus, religion can increase the risk of VAWG, while simultaneously being a preventative tool against it.

Prior research illuminates the complex role of religion in VAWG. For example, one study found that Christian women in Ghana were more likely to experience higher levels of IPV than non-Christians [4]. On the other hand, a US study illustrated that greater religious participation was associated with an 8% decrease in the likelihood of IPV victimization [16]. These discordant effects of religion on gender marginalization also provide an opening to using religion as part of the solution. This illuminates the potential in a scriptural (a religious text, such as the Christian Bible) based approach to resisting VAWG by leveraging reinterpretations of religious texts [6].

Thus, collaboration with faith-leaders is a possible tool for changing rhetoric about gender and VAWG [6,17,18]. In places where religion constitutes a core feature of a culture; a rights-based discourse is not sufficient to address the root causes of VAWG [18]. Incorporating religion and culture into VAWG interventions can help move beyond the dominant Western-based white feminist narrative of preventing VAWG [18].

## 1.3  Religion and VAWG in Ethiopia

In Ethiopia, 98% of people report a religious identification, with 27% identifying as Protestant/Evangelical (30 million people) [19,20]. Along with a highly religious population, Ethiopia also reports a high prevalence of VAWG. According to the Ethiopian Demographic and Health Survey (2019), approximately 28% of women and girls per year report experiencing IPV, and 34% of ever-married women report experiencing IPV in their lifetime [19]. Recently, Ethiopia has made progress towards gender equality, which is illustrated by having more women in government, outlawing child marriages and female genital mutilation, and creating laws protecting against gender-based violence [21]. However, gender inequality and VAWG persist. Thus, additional measures are needed to make cultural, community, relationship, and individual level changes.

One strategy is that faith-leaders could be influential change agents to address the norms and gender roles impacting VAWG [14]. Change agents are power holders who have influence over community norms [22]. Faith-leaders play important roles in the community giving support, providing resources, and creating social cohesion through the promotion of common beliefs and ethics [6,14].

The process of including religion in VAWG prevention efforts in Ethiopia is ongoing. UN Women instituted a 2020 program where over 1000 women in existing faith structures were trained to create peace committees to generate a dialogue around faith, women, and violence [20,23]. Over a thousand women received training, then generated discussions in their communities, and managed to convince many congregations to include issues of sexual violence

and abuse in their programs [20]. Additionally, in Ethiopia, researchers supported Orthodox Tewahedo faith-leaders in Tigray (Northern Ethiopia) to support capacity to address domestic violence [24]. An evaluation showed that the workshops were perceived as useful, and faith-leaders were more prepared to talk about domestic violence [24].

Despite the promising signs of faith-leaders' involvement in VAWG interventions [6,17,18], limited evidence exists on Evangelical faith-leaders' perspectives on gender and VAWG, which constitutes a large and growing number of followers, particularly in central and Southern Ethiopia [25]. The study's aim is, therefore, to add to the context-specific understanding of Evangelical faith-leaders' perceptions of gender equality and VAWG, and their views on how they can address VAWG.

## 2 Methods

### 2.1 Study design

This is a qualitative study with a constructionist perspective, drawing on in-depth interviews and data from observations from a participatory-action research workshop (more details on this workshop below). The project is embedded in a wider implementation science project. The project is a collaboration between The Ethiopian Graduate School of Theology (EGST) and Karolinska Institutet (KI) and the aim is to work with faith-leaders to address VAWG in selected areas in Ethiopia. The project was implemented in Woliso, Ethiopia, and consists of three parts: Formative qualitative research to understand faith-leaders perspectives regarding VAWG and their role in addressing VAWG (this study); integration of that knowledge in a series of faith-leader-led interventions to address VAWG in their communities (study 2) [26]; and an evaluation of the effects of the intervention on VAWG (study 3). This study presents the formative study with the aim to understand Evangelical faith-leaders' perspectives of VAWG and gender norms, as well as their views on their role in addressing VAWG in communities in Woliso, Ethiopia.

### 2.2 Study setting

The study took place in Woliso town in the Oromia regional state of Ethiopia, which is southwest of Addis Ababa. Woliso has a significant Evangelical population (18.6%), and Oromia is the region with the highest rates of VAWG in Ethiopia (38% of women reported physical, sexual, or emotional abuse) [19,27]. Amharic and Oromo are the spoken languages [19]. EGST was well suited to conduct this project in Woliso as they have strong relationships with Evangelical Churches in this area.

### 2.3 Study participants

The main inclusion criteria was that participants needed to be Evangelical faith-leaders in Woliso. The study participants were faith-leaders from seven Evangelical Churches in Woliso. Each Church had an internal hierarchy of faith-leaders that typically included seven top leaders, appointed by the Church members, who are responsible for planning, implementation, and financial and logistical decision-making. These top-ranking faith-leaders participated in Workshop #1 and then these top-ranking leaders recruited the other faith-leaders for Workshop #2. It is important to note that the gender balance was not reached in the sample, as the majority of participants identified as men, because more men hold Church leadership positions.

### 2.4 Data source and collection

Data was collected from individual interviews and observations from a participatory-action research workshop. We utilized a combination of interview and observational data to

 

triangulate data sources for increased data reliability. For example, an interview on a sensitive topic could result in response bias or be subject to the Hawthorne effect. The interview data provided more in-depth and direct responses from selected faith-leaders, while the observational data contained direct reactions to the training material and other faith-leaders' opinions. Both interview and workshop data were collected at local community space.

**2.4.1 In-depth interviews.** We recruited 14 faith-leaders for individual in-depth interviews. The recruitment process took place between October 2022 and December 2022 in Woliso and utilized a snowball methodology from initial contact with top-ranking faith-leaders, who then recruited other faith-leaders to participate. Prior to conducting each interview, verbal and written consent was gathered and recorded for documentation. A semi-structured interview guide was used with questions relating to the role of women and girls in society, the role of religion in gender roles, challenges around gender equality, and their role as faith-leaders. Interviews lasted between 40 and 60 minutes. The full interview guide is available in S1 Appendix. The interviews were conducted by WB and MD in Amharic and Afan Oromo. The interviews were recorded and then transcribed and translated.

**2.4.2 Observations and workshop materials.** The second data source included data from observations and materials from two participatory action research workshops with faith-leaders. These workshops were guided by the Channels of Hope Gender (CoHG) model, which uses a series of exercises and action-based research to encourage faith-leaders to reexamine their attitudes toward gender through the lens of Christian theology [28,29]. EGST has significant experience of working with the CoHG model. The workshops were conducted in October and November 2022. One workshop with the top-level leaders (Workshop #1); and a separate workshop with faith-leaders of congregations (Workshop #2). (26,27). NM and YA took notes and conducted observations during the workshops. The workshops, led by WB and MD, included the use of Biblical texts to foster dialogue around VAWG as well as exploring options as to how faith-leaders could address VAWG in their community (S2 Appendix). For example, faith-leaders were asked to debate interpretations of scriptures in view of VAWG. The specific material we analyzed included observational notes from these discussions (we did not audio-record the workshop due to lack of consent) and flip chart materials that summarized specific activities that addressed our research question [28]. WB and MD were well suited to conduct the research due to their training in Evangelical Theology, accreditation as facilitators of the CoHG model, and prior research and project experiences.

## 2.5 Data analysis

The authors conducted a contextualized inductive reflexive thematic analysis in Nvivo [30]. The analysis was based on Braun and Clark's methodology to generate themes by assessing the data for patterns [31]. The strategy was a critical analysis with a constructionist epistemology [32]. The data meaningfulness and textual interpretations were crucial for theme creation [32]. For the inductive approach, there is no prior existing coding scheme, and the analysis is data-driven [31,33]. Interviews were transcribed verbatim and translated into English. Back translation was completed for two interviews as a reliability check. Workshop materials (observation notes and flip chart materials) were translated and prepared for analysis. Interview and workshop data were coded together. EG and SHvW did the first rounds of double-blind coding, then a codebook was created, and a second round of coding was completed. Then, the rest of the research team (AAD, AA, WB, NM, MD, and YA) was included in the discussion of the categories and the development and refinement of the themes.

## 2.6 Reflexivity

An important part of reflexive thematic analysis is the subjective nature of the analysis. As the themes, sub-themes, and codes have been inductively created, they all contain personal biases, interpersonal biases, and methodological biases [34]. EG, who led the data analysis process, is an Italian American based in Sweden at Karolinska Institutet. EG strongly condemns any form of VAWG and believes that women have the right to individual autonomy. EG's positionality as a woman from the Global North affects the research outcomes due to her role as an outsider and perceptions of gender equality. SHvW is a white woman from Europe, who is currently living in Addis Ababa and working at KI. She has an academic background in social anthropology with a focus on Africa. In an outsider position and as a senior investigator, she collaborated closely with the research team and supported the navigation of different viewpoints in the research team.

WB, an Ethiopian woman, reflects on her lived and research-based experience on VAWG and gender equality. For WB, women's autonomy means gender justice. However, women are affected by the cultural practices in the male-dominated society and women's autonomy is denied. WB feels that parts of the faith community incorrectly interpret scripture. They force the scripture to speak on their behalf, such as men have the absolute right to have sexual intercourse without their wives' will. Based on WB's cultural understanding, men as breadwinners have the right to learn, earn, get empowered, and grow their careers, but women are not allowed the same. In this case, WB reflects that this makes women vulnerable to violence because of their ascribed gender status. WB's reflections are her viewpoint and illustrate the realities for many women, but there are also positive changes happening within Ethiopia regarding gender equality. AAD, an Ethiopian man, reflects that women have been gaining respect and autonomy. EGST has Gender Equality Guidelines that help to guide their perspective as an institution and this study. Specifically, there is a principle to use theological reflection to promote gender equality.

In terms of interpersonal reflection, the team at EGST has cultural and context-specific competencies, whereas privilege hierarchies relating to paradigms of Western and white supremacy also impact team relations. This was mitigated with consistent team meetings and an open sharing and reflection of norms and values among team members. For methodological reflexivity, the team followed current good research practices, according to The Swedish Research Council (Vetenskapsrådet) [35]. The results presented here incorporate participants' original ideas and the subjectivity of the researchers.

## 3 Ethical considerations

The study follows the guiding ethical principles to do good, avoid harm, have informed consent, respect for autonomy, and justice. The team received ethical approval from The Ethiopian Society of Sociologists, Social Workers, and Anthropologists (ESSWA) Ethical review board. The research team ensured good ethical practice by obtaining written informed consent (S1 Appendix) and anonymizing all data. Data are being appropriately stored. For informed consent, all the participants freely participated and were told the nature of the study, the methodology, and information on withdrawing from the study. There was no physical, psychological, or emotional harm toward the participants. The participants received lunch, partook in the training, and benefited through additional topical knowledge gained. The research team ensured inclusivity in global research through the collaboration with KI and EGST, and EGST's long-standing and sustainable collaboration with the Evangelical community in Woliso. Additional information regarding the ethical, cultural, and scientific considerations specific to inclusivity in global research is included in the Supporting Information (S1 Checklist). The findings have been reported following the COREQ checklist (S2 Checklist).

Ethical approval was received from the Ethiopian Society of Sociologists, Social Workers, and Anthropologists with protocol number 027/2022. Received at ESSSWA's IRB Meeting No. IRB/ESSSWA/016/022. Informed consent was received from all participants. Consent form attached as part of S1 Appendix.

## 4  Results

There were 14 interviewees, 21 participants in the workshop with high-level faith-leaders (Workshop #1), and 35 participants in the workshop with other faith-leaders (Workshop #2). There were 11 men, and 3 women interviewed. The Workshop #2 had 27 men and 8 women (Table 1). The results were generated from interviews and observational data from the two workshops. The four themes are (1) differing perceptions of VAWG as a problem in the community; (2) navigating between social and religious norms as explanations of VAWG; (3) differing interpretations of biblical scripture; and (4) navigating who has a role in countering gender inequality and VAWG (Table 2).

### 4.1  Differing perceptions of VAWG as a problem in the community

The data analysis revealed a lack of agreement among the faith-leaders about the degree that VAWG is a problem in the community (Table 2). Many faith-leaders explicitly condemned various forms of VAWG, but others failed to condemn certain elements of VAWG.

### 4.2  Acknowledging VAWG as a problem in the community

Several interviewees had witnessed forms of VAWG among their neighbors, the Church, and the wider community. Acknowledging VAWG as a problem indicates an understanding that action and change are needed to prevent harm toward women and girls.

> For instance, I can tell you about my friend and her husband…He used to insult her and physically abuse by beating her. (Woman, interview #13)

Many went further and expressed their disapproval of beatings. To illustrate this point, one woman faith-leader referred to beating as "a major sin and a disgrace" (Interview #11). Some even brought up the psychological, social, and economic impacts of VAWG. Additionally, the analysis indicated that some faith-leaders understood that there are economic, social, or geographic differences between women, which impact the women's experiences of violence. Several mentioned their disapproval of the violence against young women housemaids and the treatment of women in their rural communities.

**Table 1.  Gender of Evangelical faith-leaders from interviews and workshop data analyzed from data collection in Woliso, Ethiopia, from Oct. to Nov. 2022.**

| Characteristics | | Number (n) | Percentage (%) |
|---|---|---|---|
| Genders of Interviewees | Men | 11 | 78.6 |
| | Women | 3 | 21.4 |
| Genders of Workshop #1 | Men | 21 | 100 |
| | Women | 0 | 0 |
| Genders of Workshop #2 | Men | 27 | 77.1 |
| | Women | 8 | 22.9 |

**Table 2. Themes, sub-themes, and categories generated from interviews and workshops with Evangelical faith-leaders in Woliso, Ethiopia regarding violence against women and girls (VAWG).**

| Themes | Sub-Themes | Categories |
|---|---|---|
| Differing perceptions of VAWG as a problem in the community | Acknowledging VAWG as a problem in the community | Acknowledging VAWG as a problem |
| | | Condemning abuse in all forms |
| | | Hopes for gender equality |
| | Not fully acknowledging VAWG as a problem in community | Failing to acknowledge VAWG as a problem |
| | | Accepting elements of VAWG |
| Navigating between social and religious norms as explanations for VAWG | Social or community norms as explanation for VAWG | Traditional perceptions of women and men as explanation for VAWG |
| | | Blaming women |
| | | Silencing of Women |
| | Religion or Bible as explanation of VAWG | Lack of religion as explanation |
| | | Using Bible verses to justify abuse |
| Differing interpretations or readings of scripture and teachings on gender equality and VAWG | Bible supporting gender equality | Men and women presented as equal |
| | | Bible says women are wise and capable |
| | | Positive representations of women in Bible |
| | Bible supporting gender inequality | Old Testament view of women |
| | | Man as the head of house |
| | | Women are weak |
| | | Bible supporting punishment or abuse |
| Navigating who has a role in countering gender inequality and VAWG | Individual responsibility | Role of individual in personal sphere |
| | | Role of community |
| | Institutional responsibility | Role of government |
| | | Role of church |

> We can take as an example the things we are facing in Southwest zone such as raping, house maids being exposed to violence is becoming such a big issue in our community. (Man, interview #1)

Faith-leaders understood certain groups of women to be at higher risk within their communities, especially housemaids and those in rural areas. Nonetheless, some faith-leaders expressed hope for equality.

> If there is a work-sharing option in the house between men and women, it will be better. In our community people think house chores are only given to women. But if [we] share the load with our wives it will make it easy for them. (Man, interview #14)

### 4.3 Not fully acknowledging VAWG as a problem in the community

Contrary to the above findings, some faith-leaders (although a smaller number) felt there is sufficient progress and some condoned only elements of VAWG.

> Gender-based violence is uncommon these days. Male and female are now on equal state [terms]. Concerning the type of violence, I would say, not much is seen these days, and

it is decreasing. Women's violence is not seen much, and men and women are coming to equality. (Man, interview #8)

This statement indicates that the faith-leader does not see women's violence as much of a problem anymore and, thus, society does not need further change. Partially this can be understood by the fact that beating was generally considered wrong, but the acceptability of violence blurred around deprivation of liberties, verbal abuse, and sexual rights. The faith-leaders were divided on whether God gave men authority over women, if men could punish women, and if a husband could expect sex whenever he wants it. Several participants thought it was morally correct for men to be the sole household decision maker. A faith-leader mentioned that women were only made to be helpers and, thus, must obey men's decision-making, including issues related to bodily autonomy.

I believe men should make decisions on issues such as giving birth or how many kids to have, men should decide on the issue of security, on the relationship they have with God, men also can decide on his wife's body and on the sexual activities they do. (Workshop #2)

This quote supports the idea that requiring sex from a wife is not considered a form of sexual abuse, but rather an important aspect of respecting the commandment of marriage. However, based on a small debate that transpired between the faith-leaders during the workshop discussion, it was clear that not all the faith-leaders agreed with this, illuminating the disparity between opinions.

### 4.4  Navigating between social and religious norms as explanations for VAWG

Generally, more faith-leaders viewed VAWG as a tradition- or community-based problem rather than a religious one, yet religion remained an influential component in the discourse.

### 4.5  Social or community norms as explanations for VAWG

Gender roles and traditional norms and values were repeatedly given as explanations for VAWG. Some traditional roles expressed included the notion that women were emotional, weak, and should remain in the home, while men should be brave, fight, and hold positions of power. The reasons for VAWG, according to many faith-leaders, were not because of religious scripture, but due to culture and nature. Gender norms were given as a rationale for why men perpetrated various forms of marginalization:

In some part of our community there are still traditions that are backward. For instance, there are women who are still treated as inferior in the house, women who also work as maids are exposed to violence such as beating…also women who get raped in our community. (Man, interview #3)

Women failing to fulfill their social gender role were considered to blame in the event of VAWG. For example:

When a man sees women wearing cloth that reveals her body, he will get sexual. When he sees her on the street, he knows that he could not do anything to her so he will go back to his neighborhood and rape the young women who cannot speak up for themselves. (Workshop #2)

### 4.6  Religion or Bible as an explanation of VAWG

Faith-leaders shared differing views. They used either religious or social norms as explanations for VAWG in contradicting manners. Failing to follow God was a reason for VAWG, as this response below typifies.

> [Beating], according to God's word, is unacceptable. In our Church community, if the ladies fulfill their roles as women and the males fulfill their roles as men, I don't see any reason for conflict… (Man, Interview #9)

A failure to fulfill their respective roles as husband and wife, as determined by God, was seen by some as a reason for marital distress and an explanation for VAWG. An extension of this idea was concerns raised about the high divorce rates. One female faith-leader who provided social explanations for VAWG then proceeded to say:

> I will advise her on how to save her marriage based on the word of God and what is expected of her. For example, if she tells me he spends time with other women, I will ask her what he is missing from you and how this gap is created. (Woman, interview #11)

This contradictory explanation illuminates the difficulties in separating social and religious norms. Other faith-leaders utilized scriptural evidence to support elements of VAWG. Some of the faith-leaders claimed that the Bible discussed women as weak or unwise to justify patriarchal ideas. They also utilized biblical teachings to justify forms of verbal abuse and sexual abuse. Agreeing that men can punish those out of line, one faith-leader states:

> I believe it's biblical for a man to lead his house. This is given by God himself. As punishment, he can use anger … (Workshop #1)

### 4.7  Differing interpretations of scripture and teachings on gender equality and VAWG

Interpretations of scripture and biblical teaching were a further point of contestation. Many faith-leaders interpreted scripture with a gender equal view. Simultaneously, some interpreted the Bible with a patriarchal lens.

### 4.8  Bible supporting gender equality

Many faith-leaders expressed that men and women were created equal in the eyes of God. This is part of the justification for social norms to explain abuse rather than religious norms.

> The Bible doesn't give them [women] the inferiority that is being given to them now. They [women] are created equally like the man; the Bible shows us that they are created with full potential as a man, that they are full human beings… (Man, interview #3)

In the quote below, one faith-leader references the story of Abigal, who is described in the Bible as intelligent and beautiful, and how she prevented David from committing slaughter.

> We see women in the Bible stop what is wrong, like Abigal. [She] stopped David from doing what is not right…The society tends to think women don't have any roles in the churches but that is not true. (Man, Interview #5)

This moral depiction of Abigal provides biblical support for the equal role of women in Church and society. Several also interpreted the Bible to support the notion that women are wise and capable of leadership and decision-making.

[The] Bible speaks that a women's value is [more] precious than jewels and on other place it says that a wife is a crown for her husband…We can see how important women are from this. The Bible also speaks how wise women are and when God created us, he has made us wise. (Woman, interview #13)

This contrasts those who stated that women were not wise and required a man to make decisions.

### 4.9  Bible supporting gender inequality

Even though the same version of the Christian Bible was discussed, the faith-leaders had different interpretations of it. Some used aspects of scripture to explain the superiority of men. The New Testament was interpreted as conferring more equality but still had gender unequal aspects.

…when it comes to equality between male and female, especially in the New Testament, they are equal, but this was not the case in the Old Testament. Females were not even counted as human beings at all. Similarly, when Jesus fed 5000 people in the New Testament, women and children were not counted. (Man, interview #8)

Gender inequalities were justified because of the word of God. On multiple occasions, participants expressed that men were the head of the house and thus held decision-making power. Biblical interpretations were used to justify their beliefs, as illustrated in this statement:

I think a husband is the head of his wife. He should make the decisions…Because Eve was duped by the serpent; her husband should make major decisions. Big decisions like buying a house should be made by a husband. (Workshop #2)

Scripture was used to justify the inequalities existing between a husband and a wife. Some also found negative representations of women to justify male authority.

In Main Issues, for example, Sara led her husband astray; Job's wife instructed her husband to insult God; thus, it is not good to give women freedom in the case of salvation (in relation to God) and in deciding the number of children they have. Because he has authority over his wife because he is the primary in his house. (Workshop #2)

### 4.10  Navigating who has a role in countering gender inequality and VAWG

The fourth theme revealed the different responsible actors. Generally, the analysis illustrated that VAWG was viewed as a complex problem that required further discussion in the community, Church, and government for prevention efforts.

### 4.11  Individual responsibility

Some faith-leaders expressed a degree of personal responsibility in addressing VAWG via their own familial or marital relationships. For example:

when we see Jesus, he loved the Church by undermining [humbling] himself. Imagine a husband who loves his wife like this…I would like to ask my wife for an apology [forgiveness] because the training has challenged me. (Workshop #2)

In this quote, a faith-leader expresses that, because of the participatory action research workshop, he would like to request forgiveness from his wife for prior disrespectful or harmful

actions he has taken. He is taking ownership of his actions and demonstrating a desire to change his behavior due to the workshop. Faith-leaders also felt there was a community responsibility to address gender inequalities. Partially, this relates to the notion that VAWG is a socially, rather than religiously positioned problem.

> These [community norms] are the type of things we have to get rid of. Getting rid of these things is the responsibility of me, you and we have to teach and change our community on this issue [VAWG]. (Man, interview #1)

This faith-leader is expressing a clear desire to take action, teach, and generate community-level change regarding VAWG.

### 4.12 Institutional responsibility

Faith-leaders believed that the Church holds an additional responsibility to change the discourse around gender equality and VAWG.

> For me as a leader of the Church, I have a responsibility to teach…In addition to teaching in our Churches we have the responsibility to create awareness for the whole community… We also have the responsibility to guide, [and] counsel…these women [who have been affected by VAWG]. (Man, interview #7)

Within their capacity as faith-leaders and through the influence of the Church, they noted an additional level of responsibility to create change. Faith-leaders also expressed that the national government should continue its efforts towards better gender equality to support VAWG prevention.

## 5 Discussion

The data analysis illustrated that there is heterogeneity in the Ethiopian Evangelical faith-leaders' perceptions of VAWG. Many condemned various forms of VAWG, expressed hopes for gender equality, had an interest in working towards change, and believed they should play an important role in addressing VAWG. However, others failed to acknowledge elements of VAWG and used biblical scripture to justify forms of VAWG.

### 5.1 Heterogeneity in faith-leaders' perceptions

The heterogeneity among Ethiopian Evangelical faith-leaders' perceptions in this study demonstrates that other factors, beyond religion, are influential in determining perceptions of VAWG. This is consistent with other literature [6,36–40]. A study in the Democratic Republic of the Congo and South Africa highlighted diverse and conflicting ideas of Christian scripture regarding gender and VAWG [6]. Not all Evangelical faith-leaders have uniform ideas, even if they are from the same religion and geographic region. Other factors likely impact their perceptions, such as their Church affiliation, gender, personal history, and social norms [14,37]. These factors likely influence the interpretation and understanding of the scripture itself, which holds many complex and contradictory ideas within, and consequently, affects faith-leaders' perceptions.

Emblematic of this variation is the different understandings of which gender unequal actions constitute abuse. Ethiopian Evangelical faith-leaders in this study disagreed on whether limiting female decision-making or sexual submission are considered forms of VAWG. Similarly, a study of community stakeholders in Ethiopia found that IPV was

considered problematic, but there was a lack of awareness of the consequences of VAWG [41]. In a study among Pakistani Islamic leaders, several expressed that some forms of symbolic or mild violence were acceptable, but none supported honor killings or mutilation [42]. A complete understanding of VAWG, its causes, and its implications is still lacking [14].

## 5.2  Importance of clarity on the definition of VAWG

Clarifying the definitions and understanding of what constitutes VAWG is important to ensure comprehensive prevention efforts. Without ensuring a gender equal perspective on VAWG, Evangelical faith-leaders risk perpetuating harmful gender practices [16]. Most importantly, explicit acceptance of verbal abuse, sexual abuse in a marital context, and deprivations of liberty need to be addressed. A study of community members in the Democratic Republic of the Congo regarding spousal abuse found that marital rape was not condemned [43]. A similar study found that IPV was accepted by a sample of men and women in Western Ethiopia [44]. The training process gave the Evangelical faith-leaders a broader understanding of harm and VAWG [17,18,37,45].

Additionally, the analysis revealed that some faith-leaders in this study, including female faith-leaders, held patriarchal ideas of gender. Although not overtly harmful to women on their own, these ideas of patriarchy are influential [46,47]. There were also some indications that the responsibility or blame for VAWG was being placed on the women themselves for the way they dress or for not fulfilling certain marital roles. This allows for blame shifting away from perpetrators and societal norms overall. Thus, one main challenge for the involvement of faith-leaders in VAWG is that the patriarchal norms found within these communities facilitate and justify gender unequal power dynamics [48,49].

Overcoming the imbedded societal and religious patriarchal nature requires a change in knowledge [2,7]. In line with other recommendations, this study finds that Evangelical faith-leaders require training before being useful change agents [38,50]. One strategy that has shown some success in improving the understanding of VAWG is a scriptural approach done through re-interpreting or providing an alternative biblical explanation for gender equality [6,29].

## 5.3  Linkage between religious and cultural norms

The current study also affirms that the interconnected nature between a community's social and religious norms should be emphasized. Generally, faith-leaders placed blame for VAWG on traditional/cultural values rather than religious norms. An artificial distinction between cultural and religious norms can undermine the role of religion. Several other studies have shown that religion can create or enforce structures based on patriarchal ideas, justify practices such as female genital mutation, and perpetuate silence around VAWG [4,38,48,50,51]. Separating religion, culture, and gender socialization is an unrealistic ideal that allows for blame shifting away from religious actors [7]. Formulating gender inequalities and VAWG as consequences of intertwining religious and social norms could remove these binary realities [39]. Rather than blame shifting to other aspects of society, thinking about the role of religion and society in perpetuating VAWG can further illuminate the positionality of the faith-leaders to counter gender marginalization and VAWG.

## 5.4  Potential impact of faith-leaders as change agents

The results presented here indicate that Ethiopian Evangelical faith-leaders' have the interest and potential to become active change agents for VAWG prevention. The faith-leaders' willingness to engage with issues around gender equality and VAWG supports this potential.

As respected community members of a growing Evangelical population, faith-leaders play a pivotal role in VAWG prevention efforts. From the results presented in this study, we can see that the faith-leaders are willing to engage in a discourse around VAWG and see themselves as able to teach and influence their Church and wider communities. Further research is needed to assess the impact of VAWG and gender training on the faith-leaders.

Other studies have shown success with faith-leader-based interventions to address VAWG. For example, in a (51) three-year intervention with the faith community and leaders in the Democratic Republic of the Congo, the results showed improved attitudes towards gender equality and less tolerance for IPV post-intervention in the wider community [45]. Another intervention focused on premarital and couples counselling by faith-leaders in Uganda found that by giving a more progressive interpretation of gender roles there was a shift in power towards women and a reduction in IPV prevalence [17]. Also, a case study that explored gender training among interdenominational faith-leaders in Uganda found that they gained the skills to become advocates, held improved ideas of gender equality, and impacted perceptions VAWG [52]. Based on the initial perceptions and baseline interest from our study, the Evangelical faith-leaders in this study could have similar success to these prior interventions.

However, it is important to note that an expansion of the training and long-term support for the faith-leaders and religious communities is an important aspect of effective prevention efforts [50,53,54]. In combination with training and guidance, incorporating faith-leaders as integral stakeholders in a comprehensive approach to VAWG prevention could have a noticeable effect on rates of VAWG [54].

This research has broader implications for the involvement of faith-leaders in wider VAWG prevention efforts. Faith-leaders have been recognized as a useful, grassroots, and culturally competent group [14,55,56]. Thus, it is important for donors and stakeholders to participate with faith-leader based interventions. However, the patriarchal viewpoint of some faith-leaders, their limited understanding of VAWG, and the distinctions made between religious and social norms presented in this study indicates that interventions that work with faith-leaders must be done carefully. This suggests that future projects should consider having accountability measures, advocating for long-term adaptive programs, not collaborating with groups that promote negative gender-based stereotypes, and considering the complexities of local health system landscapes when engaging with faith-based interventions [14,57,58].

## 5.5 Strengths and limitations

This study has several strengths. This study provides a useful, context specific perspective on the perspectives and role of faith-leaders in VAWG prevention. As a research group based out of an Evangelical graduate theology school in Ethiopia, the research team had a high degree of authenticity, connectedness, and extensive local knowledge, which created a high level of trust. The team also involved a diversity of analytical perspectives in the project conceptualization, study design process, and analysis strategy, which enhances the trustworthiness of results.

The study design and data collection posed limitations. First, there was selection bias in the recruitment since the high-ranking faith-leaders selected the workshop participants. Additionally, the interviews and workshops were conducted by women, which may have impacted the responses given by some faith-leaders. Third, during the data collection process only the participants' gender was collected, thus the impact of other factors such as age, Church affiliation, or education level was not assessed. This data was not collected to protect participant anonymity. Fourth, during the workshop note taking process each participant response was not assigned to an individual. Fifth, the combined use of both interview and observational workshop data made data source comparability more challenging

but provided a more wholistic data set for analysis. Sixth, some meaning may have been lost or diluted via the translation. To lessen the extent of this limitation, two interviews were back translated.

## 6 Conclusion

In summary, the research presented here shows heterogeneity in perceptions among faith-leaders of gender equality and VAWG. Ethiopian Evangelical faith-leaders show the potential to act as change agents to create awareness and rhetoric change around gender equality and VAWG. However, training and support are important for ensuring a gender-equitable approach. Religion and scriptural-based norm-transformation strategies could become part of an integrated and comprehensive approach to countering VAWG in Ethiopia.

## Supporting information

**S1 Appendix. Informed consent and interview guide.**
(DOCX)

**S2 Appendix. CoHG Workshop #2 Agenda.**
(DOCX)

**S1 Checklist. Inclusivity in global research.**
(DOCX)

**S2 Checklist. COREQ Checklist.**
(PDF)

## Author contributions

**Conceptualization:** Wosene Berhanu, Addise Amado Dube, Adamu Addissie, Sibylle Herzig van Wees.

**Data curation:** Wosene Berhanu, Addise Amado Dube, Netsanet Megersa, Yoska Amenu, Mengistu Demeke.

**Formal analysis:** Elisa Gobbo, Sibylle Herzig van Wees.

**Funding acquisition:** Sibylle Herzig van Wees.

**Investigation:** Wosene Berhanu, Adamu Addissie, Sibylle Herzig van Wees.

**Methodology:** Adamu Addissie, Sibylle Herzig van Wees.

**Project administration:** Wosene Berhanu, Addise Amado Dube, Adamu Addissie, Sibylle Herzig van Wees.

**Supervision:** Adamu Addissie.

**Writing – original draft:** Elisa Gobbo, Sibylle Herzig van Wees.

**Writing – review & editing:** Wosene Berhanu, Elisa Gobbo, Addise Amado Dube, Netsanet Megersa, Yoska Amenu, Mengistu Demeke, Adamu Addissie, Sibylle Herzig van Wees.

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
