## [Decision Letter · Decision Letter 0]

4 Apr 2024

PGPH-D-23-02346

Learning from a gender transformative intervention among faith-leaders in Woliso, Ethiopia: a qualitative study

Dear Dr. Gobbo,

Thank you for submitting your manuscript to PLOS Global Public Health. After careful consideration, we feel that it has merit but does not fully meet PLOS Global Public Health’s publication criteria as it currently stands. Therefore, we invite you to submit a revised version of the manuscript that addresses the points raised during the review process.

We look forward to receiving your revised manuscript.

Kind regards,

Kathleen Rice

Academic Editor

Journal Requirements:

Additional Editor Comments (if provided):

Please note that I have acted as a reviewer for this manuscript, and you will find my comments below, under Reviewer 2.

Dear Dr. Gobbo,

I am pleased to inform you that our external reviewers have completed their review of your submission, and based on their assessment (and my own) we are requesting a Major Revision. In particular the reviewers requested some clarity around the methods and the implications of the results. We do hope that you will find these reviews helpful in revision your manuscript, and that you will resubmit your manuscript in due course. Their comments are as follows:

Reviewer 1:

This paper began as an interesting look at the role of faith leaders in changing violence against women and girls in their faith communities in a context where violence is rampant. However, the methodology described was very confusing, which led me to quickly lose enthusiasm for the paper. Furthermore, it is unclear what interventions would look like to raise awareness and promote transformational change on VAWG in this context. I am still not sure if the training was the intervention, or if the point of this research was to determine what such an intervention would look like. This paper needs major revisions starting with the methodology in order to be suitable for publication.

Introduction

• "Equality” and “equity” are used in the background interchangeably. Need consistency, and to ensure the correct term is being used.

• Bottom of page 5: have interventions for males in religious settings been introduced in Ethiopia?

• Is part of the study aim to understand how VAWG could be addressed by faith leaders, or what types of interventions would be feasible and acceptable? That seems to be missing from the study objectives but would seemingly be one of the most important pieces.

Methods

• CoHG seems to be focused on transforming more than knowledge; attitudes, norms, and behaviors of faith communities as well?

• Line 177: Should be “A later study stage involved conducting interviews…” Was the stage included in the current research being reported?

• It was unclear to me whether this was an evaluation of the intervention or formative research to create or adapt an intervention. I believe it is the latter, but the study objectives and methodology should be revised to make this abundantly clear. Was the observation of the training? I thought the interviews were to inform the training?

• What was included in the observation? How was it done? What was noted? There are no details on this element.

• How many people did the coding? How was coding reliability assessed.

• Line 235: “This was mitigated with consistent team meetings.” What sort of issues were discussed or create a point of contention among the team members? Also, what is “good research practice” for methodological reflexivity?

Results

• It seems the results are the themes that came out of the interviews. Where are the results of the observations? Perhaps this is an extension of the confusion around the methods. It is unclear which data are from observed training sessions and which are from individual interviews. It also does not seem that participants were asked specifically what role they see them having in changing the perceptions of their faith communities regarding violence against women and girls. They were willing to talk about how faith and culture are intertwined in a way that perpetuates violence, but it is not clear how they think this will change.

Discussion

• Again, it is not clear how the faith leaders would fulfill their role as change agents. They are interested in engaging on the topic of violence, but what does this research show they will need to do in order to affect change in their faith communities?

• Was the efficacy of the training reported? I missed it. Again, there is confusion around the methodology and what exactly is being assessed here. Clarity is needed.

• Line 521: Was Data analysis only conducted by one person? They may be more objective if not part of the data collection but how did you assess reliability?

• Another limitation is not knowing the baseline attitudes towards violence of the faith leader participants.

Editing is required throughout the manuscript for grammatical errors.

Reviewer 2:

Thank you for the opportunity to review this manuscript, which is generally in good shape and was interesting to read. It has many strengths, but is not, in my view, ready for publication. My concerns are as follows:

There are some problems with terminology, especially in the Introduction, in terms of the way that sex and gender are used interchangeably. “Male dominated views” are better termed “patriarchal views”, and “social and cultural factors associated with being male or female” should read “(…) associated with being men and women. I would urge the authors to be more careful about using the correct gendered (as opposed to sexed) language throughout the manuscript.

Likewise, “tradition” is a complex theoretical construct that is taken as naked fact (e.g. “traditional masculinity”). What does this mean, practically-speaking?

Page 3: “a critical factor is religion”. A critical factor in what?

P. 3: “believe in other realities.” What does this mean?

There are some methodological strengths to this paper (the reflexivity section; analytic approach). I would like a bit more information, however. You state that this is a qualitative study. Any particular kind? A generic qualitative study? An ethnographic study? This should be explained and referenced. Also, could you explain more about the sense in which it is a participatory action intervention study?

It would be helpful to summarize under the Data Source and Collection heading the kinds of topics that were explored in the interview guide. Also, who exactly carried out the interviews. I gather that the interviewers were not involved in the analysis. Why not?

The reflexivity section is great, however you state that “the team followed current good research practice [in terms of methodological reflexivity]”. Which is please explain this explicitly.

The material could be better situated within the broader context of religion in Ethiopia, especially given the authors note that religion, culture, and broader society are inseparable. While this study was carried out with Evangelical Christians, most Ethiopian Christians are Ethiopian Orthodox Christians, and Ethiopia also has a large Muslim population. Relatedly, I notice that there seems to be some “othering” going on among your participants (e.g., the participant who states that it is “unbelievers” who sometimes have problems with VAG; that its especially women in rural areas that housemaids who struggle with this). This seems important, as it pushes the issue onto others, potentially allowing study participants to avoid responsibility to address these issues in their own families, communities, congregations etc. This seems to be an important thread that is not taken it.

p. 14: a word is missing in the final sentence (blaming women who fail to dress in traditional for rape).

Another topic that is raised in the results but is missing from the analysis is the ways in which responsibility for limiting VAG is being placed on women, both by men and women. For example, the issue of women not dressing modestly leading to rape, and also the woman on page 15 who encourages women with unfaithful husbands to consider what they themselves could do differently (the implication here being that he is looking elsewhere because he is unsatisfied at home). I think this is an important analytic piece that could be pursued (this is merely a suggestion for how the piece could be strengthened; I do not consider it imperative).

I am confused by the first quote under the “Personal Responsibility” heading. Could you please explain more clearly what this quote means, and how it relates to personal responsibility?

Also, I wonder whether you could engage with a discussion of whether scripture itself is contradictory (as a secular agnostic, Christian scripture certainly seems to be contradictory to me), and what the implications of this are for interventions that mobilize scripture to combat GBV. How do people reconcile contradiction? How do you decide which sections are relevant, and which interpretations are correct?

The discussion of Definitional Clarity is good.

The Limitations section: it seems contradictory to state that an outside analyst is somehow more objective, when you have stated (correctly) in the methods section that objectivity is impossible given everyone is situated vis a vis a research context in which the meanings are themselves contested and socially constructed. I would perhaps rephrase this to explain that a diversity of analytic perspectives enhances the trustworthiness of the data.

Reviewers' comments:

Reviewer's Responses to Questions

**Comments to the Author**

1. Does this manuscript meet PLOS Global Public Health’s publication criteria ? Is the manuscript technically sound, and do the data support the conclusions? The manuscript must describe methodologically and ethically rigorous research with conclusions that are appropriately drawn based on the data presented.

Reviewer #1: Partly

Reviewer #2: Yes

2. Has the statistical analysis been performed appropriately and rigorously?

Reviewer #1: No

Reviewer #2: N/A

3. Have the authors made all data underlying the findings in their manuscript fully available (please refer to the Data Availability Statement at the start of the manuscript PDF file)?

Reviewer #1: Yes

Reviewer #2: No

4. Is the manuscript presented in an intelligible fashion and written in standard English?

Reviewer #1: Yes

Reviewer #2: Yes

5. Review Comments to the Author

Reviewer #1: This paper began as an interesting look at the role of faith leaders in changing violence against women and girls in their faith communities in a context where violence is rampant. However, the methodology described was very confusing, which led me to quickly lose enthusiasm for the paper. Furthermore, it is unclear what interventions would look like to raise awareness and promote transformational change on VAWG in this context. I am still not sure if the training was the intervention, or if the point of this research was to determine what such an intervention would look like. This paper needs major revisions starting with the methodology in order to be suitable for publication.

Introduction

• "Equality” and “equity” are used in the background interchangeably. Need consistency, and to ensure the correct term is being used.

• Bottom of page 5: have interventions for males in religious settings been introduced in Ethiopia?

• Is part of the study aim to understand how VAWG could be addressed by faith leaders, or what types of interventions would be feasible and acceptable? That seems to be missing from the study objectives but would seemingly be one of the most important pieces.

Methods

• CoHG seems to be focused on transforming more than knowledge; attitudes, norms, and behaviors of faith communities as well?

• Line 177: Should be “A later study stage involved conducting interviews…” Was the stage included in the current research being reported?

• It was unclear to me whether this was an evaluation of the intervention or formative research to create or adapt an intervention. I believe it is the latter, but the study objectives and methodology should be revised to make this abundantly clear. Was the observation of the training? I thought the interviews were to inform the training?

• What was included in the observation? How was it done? What was noted? There are no details on this element.

• How many people did the coding? How was coding reliability assessed.

• Line 235: “This was mitigated with consistent team meetings.” What sort of issues were discussed or create a point of contention among the team members? Also, what is “good research practice” for methodological reflexivity?

Results

• It seems the results are the themes that came out of the interviews. Where are the results of the observations? Perhaps this is an extension of the confusion around the methods. It is unclear which data are from observed training sessions and which are from individual interviews. It also does not seem that participants were asked specifically what role they see them having in changing the perceptions of their faith communities regarding violence against women and girls. They were willing to talk about how faith and culture are intertwined in a way that perpetuates violence, but it is not clear how they think this will change.

Discussion

• Again, it is not clear how the faith leaders would fulfill their role as change agents. They are interested in engaging on the topic of violence, but what does this research show they will need to do in order to affect change in their faith communities?

• Was the efficacy of the training reported? I missed it. Again, there is confusion around the methodology and what exactly is being assessed here. Clarity is needed.

• Line 521: Was Data analysis only conducted by one person? They may be more objective if not part of the data collection but how did you assess reliability?

• Another limitation is not knowing the baseline attitudes towards violence of the faith leader participants.

Editing is required throughout the manuscript for grammatical errors.

Reviewer #2: Thank you for the opportunity to review this manuscript, which is generally in good shape and was interesting to read. It has many strengths, but is not, in my view, ready for publication. My concerns are as follows:

There are some problems with terminology, especially in the Introduction, in terms of the way that sex and gender are used interchangeably. “Male dominated views” are better termed “patriarchal views”, and “social and cultural factors associated with being male or female” should read “(…) associated with being men and women. I would urge the authors to be more careful about using the correct gendered (as opposed to sexed) language throughout the manuscript.

Likewise, “tradition” is a complex theoretical construct that is taken as naked fact (e.g. “traditional masculinity”). What does this mean, practically-speaking?

Page 3: “a critical factor is religion”. A critical factor in what?

P. 3: “believe in other realities.” What does this mean?

There are some methodological strengths to this paper (the reflexivity section; analytic approach). I would like a bit more information, however. You state that this is a qualitative study. Any particular kind? A generic qualitative study? An ethnographic study? This should be explained and referenced. Also, could you explain more about the sense in which it is a participatory action intervention study?

It would be helpful to summarize under the Data Source and Collection heading the kinds of topics that were explored in the interview guide. Also, who exactly carried out the interviews. I gather that the interviewers were not involved in the analysis. Why not?

The reflexivity section is great, however you state that “the team followed current good research practice [in terms of methodological reflexivity]”. Which is please explain this explicitly.

The material could be better situated within the broader context of religion in Ethiopia, especially given the authors note that religion, culture, and broader society are inseparable. While this study was carried out with Evangelical Christians, most Ethiopian Christians are Ethiopian Orthodox Christians, and Ethiopia also has a large Muslim population. Relatedly, I notice that there seems to be some “othering” going on among your participants (e.g., the participant who states that it is “unbelievers” who sometimes have problems with VAG; that its especially women in rural areas that housemaids who struggle with this). This seems important, as it pushes the issue onto others, potentially allowing study participants to avoid responsibility to address these issues in their own families, communities, congregations etc. This seems to be an important thread that is not taken it.

p. 14: a word is missing in the final sentence (blaming women who fail to dress in traditional for rape).

Another topic that is raised in the results but is missing from the analysis is the ways in which responsibility for limiting VAG is being placed on women, both by men and women. For example, the issue of women not dressing modestly leading to rape, and also the woman on page 15 who encourages women with unfaithful husbands to consider what they themselves could do differently (the implication here being that he is looking elsewhere because he is unsatisfied at home). I think this is an important analytic piece that could be pursued (this is merely a suggestion for how the piece could be strengthened; I do not consider it imperative).

I am confused by the first quote under the “Personal Responsibility” heading. Could you please explain more clearly what this quote means, and how it relates to personal responsibility?

Also, I wonder whether you could engage with a discussion of whether scripture itself is contradictory (as a secular agnostic, Christian scripture certainly seems to be contradictory to me), and what the implications of this are for interventions that mobilize scripture to combat GBV. How do people reconcile contradiction? How do you decide which sections are relevant, and which interpretations are correct?

The discussion of Definitional Clarity is good.

The Limitations section: it seems contradictory to state that an outside analyst is somehow more objective, when you have stated (correctly) in the methods section that objectivity is impossible given everyone is situated vis a vis a research context in which the meanings are themselves contested and socially constructed. I would perhaps rephrase this to explain that a diversity of analytic perspectives enhances the trustworthiness of the data.

6. PLOS authors have the option to publish the peer review history of their article (what does this mean? ). If published, this will include your full peer review and any attached files.

**Do you want your identity to be public for this peer review?** For information about this choice, including consent withdrawal, please see our Privacy Policy .

Reviewer #1: No

Reviewer #2: No

---

## [Decision Letter · Decision Letter 1]

29 Aug 2024

PGPH-D-23-02346R1

Leveraging Faith-Leaders to Prevent Violence against Women and Girls: A Qualitative Study of Evangelical Faith-leaders’ Perceptions in Woliso, Ethiopia

Dear Dr. Gobbo,

Thank you for submitting your manuscript to PLOS Global Public Health. After careful consideration, we feel that it has merit but does not fully meet PLOS Global Public Health’s publication criteria as it currently stands. Therefore, we invite you to submit a revised version of the manuscript that addresses the points raised during the review process.

We look forward to receiving your revised manuscript.

Kind regards,

Jhumka Gupta

Academic Editor

Journal Requirements:

Additional Editor Comments (if provided):

This manuscript makes an important contribution to the field of IPV through focusing on the role of faith leaders. The remaining edits are as follows:

1. Ensure that sex and gender are not used interchangably

2. Provide justification on combining various forms of qualitative data versus analyzing them separtely.

Reviewers' comments:

Reviewer's Responses to Questions

**Comments to the Author**

1. If the authors have adequately addressed your comments raised in a previous round of review and you feel that this manuscript is now acceptable for publication, you may indicate that here to bypass the “Comments to the Author” section, enter your conflict of interest statement in the “Confidential to Editor” section, and submit your "Accept" recommendation.

Reviewer #2: All comments have been addressed

Reviewer #3: (No Response)

2. Does this manuscript meet PLOS Global Public Health’s publication criteria ? Is the manuscript technically sound, and do the data support the conclusions? The manuscript must describe methodologically and ethically rigorous research with conclusions that are appropriately drawn based on the data presented.

Reviewer #2: Yes

Reviewer #3: Yes

3. Has the statistical analysis been performed appropriately and rigorously?

Reviewer #2: N/A

Reviewer #3: N/A

4. Have the authors made all data underlying the findings in their manuscript fully available (please refer to the Data Availability Statement at the start of the manuscript PDF file)?

Reviewer #2: Yes

Reviewer #3: Yes

5. Is the manuscript presented in an intelligible fashion and written in standard English?

Reviewer #2: (No Response)

Reviewer #3: No

6. Review Comments to the Author

Reviewer #2: This paper is much improved from its original version, and I now have only minor suggestions/concerns. The methods section is now very good. My minor concerns are as follows:

My main concern is that there is still inappropriate conflation of gender and sex (int he sense that they are used interchangeably ins situations where there is inaccurate). Yes, I understand that the distinction is problematic/ not meaningful for many study participants, and thus in the results we would expect these concepts to be conflated. But in descriptive and/or analytic sections of the paper, using them interchangably is frankly inaccurate. For instance, on page 3-4, where it reads "in this study gender equality refers to the notion that men and women have the same access to opportunities and life changes and that they are not dependent on or contrained by their sex" should really use consistent GENDER as opposed to sex language throughout. Likewise the table on page 12 lists the "gender" of participants as being male and female, when it should be written as "men" and "women."Male and female quite simply is not their gender, so the table is inaccurate in its current form.

p. 3: in the first paragraph of the paper you state that "VAWG is defined as "....". Defined by whom? Surely there is more than one definition, and I think it makes sense to specify whose definition you are using.

p. 3: "avoidance of femininity" is a bit vague. Surely traditional masculinity can entail interacting (in specific ways) with normatively feminine women. I think this could be worded to be more clear.

p. 5: "leveraging re-interpretations." Re-interpretations of what, exactly?

p. 14 the sentence "additionally, the analysis held indications of some faith-leaders understanding economic, social, or geographic differences" is unclear to me.

p. 20: the quote which states "he loved the Church by undermining himself" is unclear, I think because "undermining" is probably not being used the way it is typically used in English. Could you clarify, perhaps in a square bracket, what is meant here?

p. 26: forth should read "fourth"

Reviewer #3: There were minor grammatical errors and a need for rephrasing, specifically on lines 93 and 140. Authors should do another thorough review for minor mistakes.

The analysis is sound, and the authors clearly outline how it took place. However, there are two sources of qualitative data: (1) interview transcripts and (2) workshop notes, which for some would be considered a mixed qualitative analysis because of the mode of data collection. It would strengthen the methods section if the authors described why they chose to combine the interview data and workshop data rather than conducting separate analyses. Questions I would recommend answering: How does the combination of the data strengthen the analysis and results? What are the limitations of combining the data?

The discussion section offers significant insights into the field, particularly the use of faith leaders in combating GBV through social-behavioral interventions. This is a commendable contribution to the field.

7. PLOS authors have the option to publish the peer review history of their article (what does this mean? ). If published, this will include your full peer review and any attached files.

**Do you want your identity to be public for this peer review?** For information about this choice, including consent withdrawal, please see our Privacy Policy .

Reviewer #2: No

Reviewer #3: No

---

## [Decision Letter · Decision Letter 2]

13 Nov 2024

PGPH-D-23-02346R2

Leveraging Faith-Leaders to Prevent Violence against Women and Girls: A Qualitative Study of Evangelical Faith-leaders’ Perceptions in Woliso, Ethiopia

Dear Dr. Gobbo,

Thank you for submitting your manuscript to PLOS Global Public Health. After careful consideration, we feel that it has merit but does not fully meet PLOS Global Public Health’s publication criteria as it currently stands. Therefore, we invite you to submit a revised version of the manuscript that addresses the points raised during the review process.

I commend you for the remarkable job you have done addressing reviewers' comments. I am willing to accept this manuscript for publication pending a few minor revisions to improve the clarity and readability of the text. I invite you to submit a revised version of the manuscript that addresses the points raised below.

I recommend using a language editing service to identify and correct a few remaining spelling and grammatical errors. For example:

-Using one of UK or US English consistently throughout the text

-Page 8, line 200: Should be “Hawthorne effect”

-Page 9, line 217: Abbreviations should be written in brackets after the first time a term is used

- Page 9, line 231: Should be “in Nvivo”

-Page 10, line 258: There should be an apostrophe at the end of “wives”

-Page 11, line 263: “that” is written twice

-Table 1: Should be “gender”

-Page 17, line 382: Should be “religious and social norms”

-Page 19, line 429: Should be “gender unequal”

-Page 25, line 566: There should be an “a” before the word useful

-Page 26, line 586: There should be an apostrophe in the word “participants”

The following changes are required for acceptance:

1. Please use the COREQ checklist, or another relevant reporting checklist, to ensure complete reporting of the methods and results. A supplementary document should be included detailing where in the manuscript the checklist items are reported.

2. The terms (in)equity and (in)equality are still used interchangeably in some parts of the manuscript. Please review these sections and revise accordingly (page 6, lines 144-146; page 18, lines 403-406 where title mentions equality but then paragraph refers to equity; )

3. Please provide additional information to clarify the following points:

Page 6, line 155: It's not clear what the percentages are referring to. Are you indicating that 13 to 30% of all women in the regions participated in the program?Page 7, line 187: Please report the rate of VAWG in OromiaPage 9, line 220: Please clarify what is meant by “high-level leader” as those not familiar with Evangelical church system in Ethiopia will not know what this meansPage 9, lines 226-227: It is indicated that the workshops were not audio-recorded but no information is provided about interviews. Please specify whether in-depth interviews were audio-recorded or not in the appropriate paragraph.Page 10, lines 236-237: Please specify how many interviews were backtranslatedPages 10-11: Please report consistent information across authors in the reflexivity section. It is not clear why only EG’s faith is reported and why WB and AAD’s ethnicity, race, or citizenship are not reported.Page 10, line 249: Please clarify how “EG’s positionality as a woman from the Global North affects the research outcomes”Page 12, line 302: Please specify which workshop (#1 or #2) is being referred toPage 16, line 361: Please clarify the specific observations that made it clear “that not all the faith-leaders agreed with this”Page 25, lines 555-557: Please specify how exactly the gender training in Uganda “impacted overall rates of VAWG”

We look forward to receiving your revised manuscript.

Kind regards,

Marilyn Naana Ahun, PhD

Academic Editor

Journal Requirements:

Additional Editor Comments (if provided):

Reviewers' comments:

Reviewer's Responses to Questions

**Comments to the Author**

1. If the authors have adequately addressed your comments raised in a previous round of review and you feel that this manuscript is now acceptable for publication, you may indicate that here to bypass the “Comments to the Author” section, enter your conflict of interest statement in the “Confidential to Editor” section, and submit your "Accept" recommendation.

Reviewer #2: (No Response)

2. Does this manuscript meet PLOS Global Public Health’s publication criteria ? Is the manuscript technically sound, and do the data support the conclusions? The manuscript must describe methodologically and ethically rigorous research with conclusions that are appropriately drawn based on the data presented.

Reviewer #2: Yes

3. Has the statistical analysis been performed appropriately and rigorously?

Reviewer #2: N/A

4. Have the authors made all data underlying the findings in their manuscript fully available (please refer to the Data Availability Statement at the start of the manuscript PDF file)?

Reviewer #2: Yes

5. Is the manuscript presented in an intelligible fashion and written in standard English?

Reviewer #2: Yes

6. Review Comments to the Author

Reviewer #2: Thank you very much for your efforts to address the concerns that I had raised during previous reviews. I now consider the manuscript publishable.

On page 9 (line 220) there are missing words (I believe that the text should read "one workshop was held with..."

7. PLOS authors have the option to publish the peer review history of their article (what does this mean? ). If published, this will include your full peer review and any attached files.

**Do you want your identity to be public for this peer review?** For information about this choice, including consent withdrawal, please see our Privacy Policy .

Reviewer #2: No

---

## [Editor Report · Decision Letter 3]

6 Jan 2025

Leveraging Faith-Leaders to Prevent Violence against Women and Girls: A Qualitative Study of Evangelical Faith-leaders’ Perceptions in Woliso, Ethiopia

PGPH-D-23-02346R3

Dear Ms Gobbo,

We are pleased to inform you that your manuscript 'Leveraging Faith-Leaders to Prevent Violence against Women and Girls: A Qualitative Study of Evangelical Faith-leaders’ Perceptions in Woliso, Ethiopia' has been provisionally accepted for publication in PLOS Global Public Health.

Best regards,

Marilyn Naana Ahun, PhD

Academic Editor